# Telemedicine as a Medical Examination Tool During the Covid-19 Emergency: The Experience of the Onco-Haematology Center of Tor Vergata Hospital in Rome

**DOI:** 10.3390/ijerph17238834

**Published:** 2020-11-27

**Authors:** Massimiliano Postorino, Michele Treglia, Jacopo Giammatteo, Margherita Pallocci, Giulia Petroni, Giuseppe Quintavalle, Ombretta Picchioni, Maria Cantonetti, Luigi Tonino Marsella

**Affiliations:** 1Department of Biomedicine and Prevention, University Tor Vergata, 00133 Rome, Italy; maxpostorino@libero.it (M.P.); michelemario@hotmail.it (M.T.); giammatteojacopo@gmail.com (J.G.); giulia.petroni17@gmail.com (G.P.); ombretta.picchioni@uniroma2.it (O.P.); cantonetti@med.uniroma2.it (M.C.); marsella@uniroma2.it (L.T.M.); 2Local Public Health Unit ASL Roma 4, Civitavecchia, 00053 Rome, Italy; giuseppe.quintavalle@aslroma4.it

**Keywords:** telemedicine, COVID-19, onco-haematology, ethics, legal medicine

## Abstract

Background: Our study analysed the outpatient activity of the onco-hematology Complex Operative Unit (UOC) of Tor Vergata Hospital, Rome coronavirus disease 2019 (Covid-19) center, where, as a result of the sudden and unexpected emergency, healthcare services were provided through telemedicine procedures that can be considered very close to Telehealth. Aim of the study: our retrospective study aimed to assess the widespread use of telemedicine in terms of feasibility and safety related to adverse events, a crucial experience which will make it possible to predict any effective use of such a method in patients with hematological disorders even after the end of the Covid-19 emergency. Materials and methods: At the Day Hospital clinic, from 8 March to 31 May 2020, an outpatient group received 3828 medical teleconsultations and 11,484 additional contacts following the first examination; each patient examined through the telematic method required an average of three supplementary contacts via e-mail or telephone. Results: The follow-up lasted 145 days, and all the events that occurred were monitored. In total, we recorded 16 clinical adverse events, 5 of which classified as major events, and 11 as minor events. Conclusion: The 3828 telematic clinical examinations and the 11,484 additional contacts following the first examination carried out by the onco-haematology UOC of Tor Vergata Hospital, proved how telemedicine, albeit in its basic form, was a key tool in facing the sanitary emergency caused by the sudden spread of Covid-19. An experience that can be considered reliable enough to be replicated in possible post-Covid-19 emergencies. From a medical forensic point of view, the main issues to consider are informed consent, personal data management and professional responsibility profiles.

## 1. Introduction

The coronavirus disease 2019 (Covid-19) emergency forced healthcare facilities and doctors to provide healthcare services through telemedicine, changing the way clinicians treat patients not infected by the virus but in the need of diagnosis, therapies and treatments for other pathologies. In order to reduce the risk of infection and with the aim of keeping social distancing, healthcare facilities and health professionals have established new inhouse medical protocols by facilitating the provision of remote healthcare services via digital devices and the Internet. In particular, healthcare facilities had to consider new procedures aiming to avoid worsening in patients suffering from chronic and oncological diseases by ensuring a continuous, regular, mandatory and undeferrable monitoring related to their specific pathology.

This mainly implied an increase in the use of the telemedicine activities.

During the Covid-19 emergency, this type of caring, already in place but never really implemented by health professionals, has become a fundamental tool in order to meet patients’ need for care. The use of telemedicine in clinical practice has been receiving increasing attention for over a decade. Considerable efforts have been made in order to harmonise and make this tool usable by health professionals and citizens. In particular, the European Commission communication (COM-2008-89) entitled “*Telemedicine for the benefit of patients, healthcare systems and society*” of 4 November, 2008, sets out a series of initiatives, addressed both to the Community and to each Member State, with the aim of encouraging the use of this tool in daily clinical practice and removing any obstacles that could affect its valid and effective implementation.

According to the National Lines of Approach provided by the Italian Ministry of Health, when we refer to telemedicine, we mean a way of providing health care by means of Information and Communication Technologies (ICT), that is to say, technological innovations through which it is possible to carry out a medical consultation although the doctor and patient are in two different locations. The Italian approach provides three different operational ways: specialist telemedicine (through televisit and teleconsultation), remote health assistance (a social assistance system concerning the management of frail or elderly people at home), telehealth (it concerns systems and services enabling a connection between patients—especially those with chronic diseases—and doctors in order to provide diagnosis, monitoring, management and increase patients’ awareness of their own condition) [1].

Our study analysed the outpatient activity of the onco-hematology Complex Operative Unit (UOC) of Tor Vergata Hospital in Rome (Rome Covid-19 center) where, as a result of the sudden and unexpected emergency, healthcare services have been provided through telemedicine procedures that can be considered very close to Telehealth. It includes Telemonitoring, with the multiple intent of understanding the health data communicated by the patient, choosing the appropriate therapy and training/providing information to the patient.

The main purpose of this retrospective assessment is to predict a concrete extended application of telemedicine in the post-Covid-19 era.

## 2. Materials and Methods

In the onco-haematology UOC of Tor Vergata Hospital in Rome from 8 March to 31 May 2020, an outpatient group received telematic health services; in total, we trated an average daily number of 66 patients (range 52–84), with an average of 13 patients per medical team (range 10–18). The outpatient group consisted of male (58%) and female (42%) patients with an average age of 58 years (range 18–93). All patients gave their consent to the processing of personal data. 

On each occasion, the patient was informed both by telephone and by e-mail of the impossibility of carrying out a face-to-face outpatient consultation and, therefore, that telemedicine methods were required. Furthermore, the ways in which personal health data were processed and the way in which they were recorded were always communicated via telephone and e-mail.

In practice, according to our telemedicine protocol, non-postponable appointments were kept on schedule and changed, as far as possible, into remote consultations; to all patients assisted, the medical doctors, on the days preceding the examination (with a notice of 1–6 days, average of 4 days) performed a first telephone screening

The first step of the protocol consisted in the collection of the following clinical data: patient’s general condition, respiratory and heart rate, blood pressure, skin and mucous membranes pallor, specific conditions caused by the disease through a self-assessment of fatigue, presence of lymphadenopathy, general or sectoral oedema and abdominal organomegalies. As a result of these data, radiological examinations and lab tests were prescribed.

The clinical self-assessment had to be transcribed by the patient and sent to the clinicians via email, together with the diagnostic tests requested during the first telephone contact. As a result of the e-mail sent by the patient, a new e-mail from the medical staff followed with the notification of a new appointment.

The patients’ telematic data were recorded in a medical chart specifying the examination method, with the note “Covid-19 period”. In the event that it was necessary to resort to a face-to-face examination, the patient was called and summoned for the next day in the out-patient clinic.

In total, 3828 telematic clinical examinations and 11,484 additional contacts following the first examination were carried out in the period considered. Each patient examined through a telematic mean required an average of three supplementary contacts via e-mail or telephone (range 2–5). 

At first, this procedure caused organisational problems due to the management of the outpatient group’s e-mails. Difficulties were overcome as soon as patients were given a doctor’s institutional e-mail address.

In order to evaluate retrospectively the suitability of remote patient management in the post-Covid-19 era, the influence of patients’ poor compliance on adverse events was examined.

The adverse events were divided into:(a)Major events (in the case that the telematic procedure did not enable the detection of the clinical problem that a de visu service would probably have highlighted);(b)Minor events (i.e., delay in the availability of home therapies due to the lack of prescriptions or due to the patient’s difficulty in managing computer technology).

## 3. Results

The follow-up taken into consideration lasted 145 days, during which all the events that occurred during the completion of this telematic procedure were monitored.

Low compliance was observed in less than 0.3% of the cases (12/3828), and in this case, a further telematic check was carried out. 

In detail (Figure 1): 6/12 patients were elderly, with caregivers outside the region; 4/12 patients lived with an elderly caregiver who had difficulties in understanding and managing the instructions provided; 2/12 patients were without caregivers and with hearing impairments which precluded an adequate communication.

With regard to Major Adverse Events, the incidence was less than 0.13% (5 out of 3828), and in no one case, they significantly affected the diagnosis or, in particular, the evolution of the disease.

The following is a description of these five individual cases.

A patient with high-grade non-Hodgkin lymphoma (NHL) in complete remission and in follow-up presented latero-cervical lymphadenopathy, which from self-examination was defined as centimetric and painful. The medical doctor, not being able to fully validate the self-assessment, prescribed an ultrasound scan that was performed after 40 days because of management issues related to the emergency. Later, contrary to what the patient stated, the ultrasound showed the pathological aspects of the lesion, therefore, the patient was summoned for an examination and underwent a lymph node biopsy which confirmed cancer recurrence. However, this diagnostic delay did not significantly affect the treatment and therapeutic response of the patient.An elderly patient with NHL did not pay attention to an intermittent edema of her lower limbs during the interview; only later, because of the constant and massive appearance of this clinical symptom, did she contact the health assistance center again and, after being immediately summoned, cardiological examinations showed a condition of heart failure of medium severity due to dilated cardiomyopathy. At present, the patient has good haemodynamic compensation.A patient with chronic lymphocytic leukemia (CLL) did not undergo the blood tests requested several times by email, due to the logistical inability to perform the tests; after missing two telematic interview appointments, the doctors called him to perform the haematological tests in situ, which showed a worsening of his disease. This diagnostic delay did not cause any damage, and the patient is under “watch and wait”.A patient with haemolytic anemia under a progressive steroid dose reduction did not show bone symptoms related to cortisone treatment (intermittent lumbar pains) in the telematic interview, and this aggravated his vertebral osteopaenia resulting in an L5 fracture. In this case, the diagnostic delay resulted in physical damage, which was amended following kyphoplasty treatment.A febrile, neutropenic patient with respiratory distress was treated by her family doctor, with the suspicion of Covid-19 infection, without consulting the haematologist in charge; it later emerged that the patient presented bacterial pneumonia which was diagnosed with a delay of about 5 days and finally resolved with immediate hospitalisation after telephone contact with the haematology unit.

Minor Events, which were also without serious outcomes, had an incidence of 0.29% (11 out of 3828), mainly attributable to administrative/technical problems that required a de visu examination (Figure 2):6/11 patients did not access therapies because they were unable to obtain a medical prescription from the family doctor.5/11 patients’ delays in using the Telehealth service provided were related to the domestic management of ICTs.

## 4. Discussion

In 2010, the European Commission defined the concept of telemedicine related to the management of ICTs with the aim of encouraging the use of this tool in each Member State [2]. In June 2012, the Italian Ministry of Health set out the extended use of telemedicine, and in 2019, Ministry Directives pointed out the importance of mapping the use of telemedicine at a national level and defining its fields of application and guidelines. This project is still ongoing. 

Nevertheless, despite the efforts made in the last decade, the overall Italian situation during the period of the Covid-19 pandemic, has appeared to be fragmentary [3] and partly ineffective.

Several healthcare structures, including Tor Vergata General Hospital in Rome, experienced the pressing need to devise alternative forms of healthcare assistance, where applicable, in order to protect patients and caregivers by limiting their mobility and so their potential exposure to the infection.

When the pandemic broke out, the healthcare facilities experienced the need for the continuity of care through the use of effective, safe and practicable tools and protocols [4]. 

Although it is not surprising that, in such context, the use of telemedicine was clearly the only option for healthcare structures and professionals despite the lack of its previous systematisation in the Italian National Healthcare System (a shortcoming not directly attributable to single facilities or caregivers), possible medico-legal issues with a great ethical and deontological impact must always be taken into account.

At the beginning of our investigation following the pandemic situation, we had to deal with completely new ethical, deontological and medico-legal challenges related to the doctor–patient relationship. The main issues concerned patients’ remote management, the reliability of patients’ self-assessment and the availability of medical doctors, hospital equipment as well as appropriate IT devices for the patients. A last but not least consideration concerns the average age of the patients (over 58 years), implying a low compliance with the management of the provided information and instructions. [5]

The use of telemedicine may entail two main risks:the unsustainability of the huge amount of clinical data to analyse in an acceptable period of time and then compare and record in medical charts in order to plan the related diagnostic, therapeutic follow-up;the impossibility of verifying the reliability of the patient self-assessment, with risk of diagnostic and therapeutic delays due to an underestimation of the symptoms by the patients.

The comprehensive management of the 3828 clinical examinations and the 11,484 additional contacts carried out over the period from March to May required the involvement of 10 clinicians per day (range 8–14), with a daily activity of 7.5 h (range 5–12) per clinician. This was crucial to provide an appropriate response in less than 5 days (range 1–5), managing at the same time any sudden emergency. 

This complicated organizational system has allowed to set the incidence of adverse events below the threshold of 0.05%, an outcome that can be considered very close to the usual clinical, diagnostic and therapeutic errors rate in the daily de visu medical practice. These outcomes show that no major events led to irreversible damage to the patients. Minor events due to technical and management issues were communicated to the Health Authority in order to be prevented in the future.

Consequently, national guidelines or in-house protocols should be established, standardising the procedure to select patients who can undergo healthcare assistance through telemedicine methods.

These protocols must necessarily take into account:the clinical aspects of the patient: age, compliance, autonomythe aspects of the disease: acute or chronic stage, the need of chemotherapy, a minimum of de visu checkupsthe presence of caregivers and the availability of adequate IT tools.

Another aspect not to be underestimated is the clinicians’ IT equipment and their training, since they need a specific training on how to use telemedicine and its potential. Furthermore, the Health National System should provide healthcare facilities, clinicians and, where appropriate, basic IT devices to patients, in order to provide healthcare services in complete safety.

In order to make such service feasible, also with regard to the professionals directly involved in providing it, the healthcare structures have to equip the professionals with both appropriate telemedicine electronic devices (e.g., in the case of teleconsulting via webcam, a laptop or desktop rather than a smartphone) and suitable rooms, neutral in style and orderly, providing adequate privacy [6].

Continuing on the subject and investigation of possible medico-legal issues, we agree with a recent review of legal literature highlighting how the most frequent problems concern informed consent to undergo a medical procedure, the processing and protection of sensitive data and professional liability [7].

With regard to informed consent, the main regulations on the subject are provided by Law No. 219/2017. Clearly, even when medical assistance is supplied within the frame of telemedicine and telehealth, patients must be adequately informed and made aware of all aspects of the medical procedure by which health services are to be delivered as well as of all its limitations. As set out by the mentioned law, informed consent must be evidenced in documentary form or by audio–video recordings and must be included in the patient’s medical record. It is advisable for healthcare structures to use an informed consent form specifically devised for patients receiving telemedicine healthcare [8,9].

A further issue to explore concerns the processing and retention of sensitive data exchanged between doctors and patients or between healthcare professionals during a teleconsultation, since confidentiality on data concerning the health of citizens who access state or private healthcare facilities must be ensured at all times.

At a European level, the reference rules on personal data protection are contained in Regulation 2016/679, the “General Data Protection Regulation” (GDPR) [10].

In ordinary circumstances of healthcare provision, there is no need on the part of the clinician to submit a privacy statement to the patients for the processing of personal data on health, given that health professionals are subject to the obligation of secrecy [11]. 

By contrast, guidelines issued in 2012 on the handling and processing of sensitive data when delivering telemedicine services are stricter. They establish the need to submit a privacy statement concerning the processing of personal health data to patients and the rights of the patients.

Information on the processing of personal data is usually given in writing or electronically through other devices and only in special cases, when a person’s identity is known, it may be provided orally. According to the Italian Data Protection Authority (Garante della privacy GDPR), in Italian) “*the Data Controller has the task to choose the best case-specific option taking into account all the issues involved in data processing and the circumstances in which it occurs*.” Also, with reference to the privacy statement on processing personal health data when delivering telemedicine services, there is no obligation for it to be in writing, though that seems to be the most recommended form, as it can prove the patient’s/data subject’s explicit consent [12].

Therefore, with the aim of delivering telemedicine services effectively and avoiding legal issues on privacy, every healthcare structure or operational unit involved is required to appoint a data controller and at least a data protection officer (if applicable). Moreover, patients must be informed on how their data shall be managed and stored, who shall have access to it and, overall, what their rights are concerning such data. In addition, healthcare structures must ensure that the “*transfer of all patients’ files with voice, video, images recordings is encrypted and complies with the existing regulation on privacy and security*” [5].

The GDPR provides for adequate safety measures in relation to the data controller who is under a means-related obligation (not under an obligation to achieve a result), ensuring that they are reasonably effective with reference to existing knowledge and procedure [13].

In light of the above, it is doubtless that health structures are also required to make a considerable, yet necessary, effort in terms of organization and management.

Lastly, another issue to examine is that of health professional liability ensuing from malpractice, which some define as telenegligence [7].

Within the context of teleconsulting, there are obvious limitations linked to the very methodology (e.g., the inability to carry out a full objective examination) that could interfere with its effectiveness. That is the reason why eligible patients should be chosen carefully, deferring to in-person patient encounters all unclear or difficult cases, so as to reduce the risk of treatment and/or diagnostic blunders.

As far as legal issues are concerned, medical procedures carried out within the frame of telemedicine services ought to fall within provisions presently laid down in Italy by Law Gelli Bianco, if they are to be conceived as optional, yet equal to, conventional healthcare services [14].

At any rate, healthcare clinicians wishing to deliver telemedicine services need effective guidelines or in-house protocols valid within their work structure, outlining best practices to provide telemedicine. Such guidelines should be drawn on the basis of and address the specific context in which clinicians are to provide healthcare, so as to help them assess whether a patient is eligible for teleconsulting.

## 5. Conclusions

The 3828 telematic clinical examinations and the 11,484 additional contacts following the first examination carried out by the onco-haematology UOC of Tor Vergata General Hospital in Rome, proved how telemedicine, albeit in its basic form, has been a key tool in facing the sanitary emergency caused by the sudden spread of Covid-19.

Despite the emergency, the respect of the Privacy Regulation and the appropriate provision of informed consent were guaranteed, and possible legal challenges resulting from malpractice cases due to some difficult-to-manage situations were avoided.

With regard to privacy, all the information required by the GDPR as well as the consent to the processing and transfer of personal health data were communicated in verbal form. At present, no dispute chargeable to this procedure has been addressed to health professionals by their patients. 

After a follow-up that lasted over 185 days and the reopening of the outpatients’ department, patients have come back for checkups, and no serious adverse events were found.

To conclude, the experience of the onco-haematology UOC of Tor Vergata General Hospital in Rome highlights that the improvement of telemedicine in daily medical practice is possible, providing that appropriate health facilities are built, and specific protocols on the organization and management of patients are drawn up.

A last aspect that cannot be underestimated and that is related to the need of writing guidelines and protocols on medical services provided through telemedicine, is the risk for clinicians and healthcare facilities of incurring into legal malpractice lawsuits. At present, in fact, the lack of guidelines and best practice rules precludes appropriate patient management. In particular, the assessment of good medical practice for physicians who, using telemedicine, could incur into legal challenges may result difficult and discretional.

## Figures and Tables

**Figure 1 ijerph-17-08834-f001:**
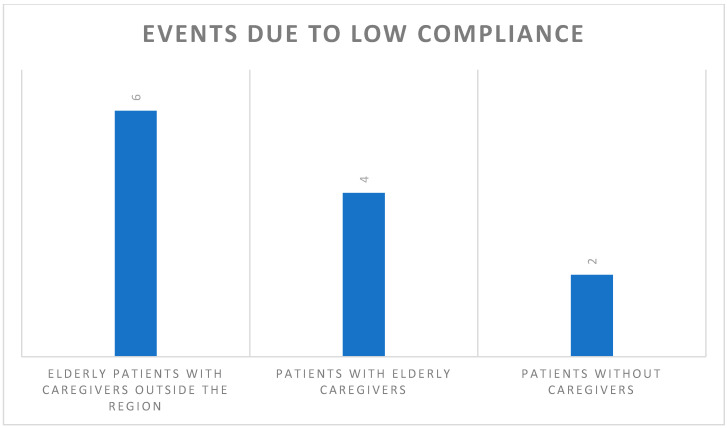
The total number of events due to low compliance was 12/3828.

**Figure 2 ijerph-17-08834-f002:**
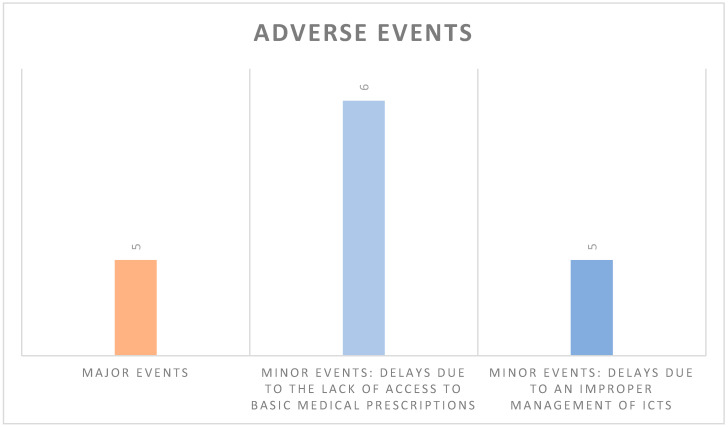
The total number of adverse events was 16/3828, 5 of which classified as major events and 11 as minor events.

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
