# Peer review of "Telemedicine as a Medical Examination Tool During the Covid-19 Emergency: The Experience of the Onco-Haematology Center of Tor Vergata Hospital in Rome"

_ijerph, 2020, doi:10.3390/ijerph17238834_

Round 1
Reviewer 1 Report
Overall
- Although well written, the Authors should have in mind that a single paragraph means a single topic. Here, the manuscript contains several mini-paragraphs, many of them as a single sentence, that should be merged into more comprehensive paragraphs. Therefore, all manuscript must be re-arranged accordingly. As mere examples, I indicated how to proceed for the introduction section. Please amend the rest of the manuscript.
Abstract
- Line 13: Please define UOC
- Line 16-17: this sentence should be deleted
Introduction
Line 40-42: these lines should be part of the previous paragraph and not stand alone
Line 60-73: again, these lines should constitute a single, and not 3 different paragraphs
Line 74-83: again, these lines should be merged into a single paragraph
The introduction should provide a clearer rationale and a so-what for this study
Methods
No statistical analysis section is described. As such, I don’t know what data were recorded and how these were analyzed. As such, the methods and results sections are not consistent.
Discussion
The discussion is unnecessarily long. Together with its rearrangement as previously indicated, I suggest shortening it to more straightforwardly indicate the results of the study, explanations for that, and how to use the novelty contained in the manuscript. Very same for the conclusion.
Author Response
Please see the attachment.
Point 1: Although well written, the Authors should have in mind that a single paragraph means a single topic. Here, the manuscript contains several mini-paragraphs, many of them as a single sentence, that should be merged into more comprehensive paragraphs. Therefore, all manuscript must be re-arranged accordingly. As mere examples, I indicated how to proceed for the introduction section. Please amend the rest of the manuscript.
Abstract
Line 13: Please define UOC
Line 16-17: this sentence should be deleted
Introduction
Line 40-42: these lines should be part of the previous paragraph and not stand alone
Line 60-73: again, these lines should constitute a single, and not 3 different paragraphs
Line 74-83: again, these lines should be merged into a single paragraph
Response 1: the authors agree with the observations in point one and therefore all the suggestions have been summarised and accepted; the whole text has been rearranged following the suggested conceptual line in order to make it easier and more immediate to read.
Point 2: The introduction should provide a clearer rationale and a so-what for this study.
Response 2: In the opening session we focused more on the background in which the clinical experience reported was developed; this is because, since the work was carried out retrospectively, it was considered correct to provide the reader with a background as complete as possible. However, since the observation made is relevant, the introduction has been integrated with a clear definition of what the proposed analysis aims to achieve, as follows: “In this retrospective study we want to evaluate the feasibility and safety (in terms of adverse events) of an extended use of telemedicine: from this experience it will be possible to understand if it is applicable and effective in haematological patients even after the emergency”.
Point 3:
Methods
No statistical analysis section is described. As such, I don’t know what data were recorded and how these were analyzed. As such, the methods and results sections are not consistent.
Response 3: In a retrospective analytical study, but not a comparison study, it is not possible to carry out a statistic that gives significance to the work, only a description of the experience is possible in order to draw indications of prospective meaning (what represents the aim of this study). The relative evaluation of the single clinical parameters, as well as the statistical evaluation of the same, is beyond the scope of the study, therefore the method intended as a telemedicine protocol must be compared with the results of its feasibility and safety.
All this considered, even though the observation in point 3 is reasonable, it is not possible to integrate it into the text.
Point 4:
Discussion
The discussion is unnecessarily long. Together with its rearrangement as previously indicated, I suggest shortening it to more straightforwardly indicate the results of the study, explanations for that, and how to use the novelty contained in the manuscript. Very same for the conclusion.
Response 4: The discussion was reduced and the data were evaluated in terms of safety and feasibility of telemedicine, as well as the conclusion.

Reviewer 2 Report
Summary of article (Date: Nov 11th, 2020)
The authors have investigated 3,828 telematic clinical examinations and 11,484 additional contacts during the COVID-19 crisis. To the best of my knowledge, this paper has examined a huge number of contacts in telemedicine.
Comments from a reviewer (Date: Nov 15th, 2020)
It is important for our society to investigate the weaknesses or adverse effects of telemedicine during the COVID-19 crisis. This interesting paper focuses on over 3,000 telematic clinical examinations in an Italian hospital. I would like to congratulate the authors on their efforts.
Here are my comments and suggestions about the manuscript.
[1] “Abstract” and “Introduction”: While the authors provide a detailed description of telemedicine, the hypotheses in this study are not clear. This makes it a little difficult for the reader to understand why this study was conducted. I would recommend that the authors provide a clear notation of what the hypothesis of this study is in the Abstract and Introduction.
[2] “Result”: I would recommend spelling out what “NHL” and “CLL” stand for.
[3] “Discussion”: The authors discussed general cautions and other aspects of telemedicine. However, I believe that the discussion should focus upon what was done in this study and its results.
I found the results of this study itself remarkably interesting. I think it's worth discussing the fact that two of the 12 people who were low compliant had hearing problems.
I think that is very interesting because while some of the previous papers have suggested that telemedicine may be difficult for people with hearing and vision problems, we don't have enough concrete data yet. I encourage authors to discuss these results in the discussion part. Please refer to the following papers:
DOI: 10.1001/jamaneurol.2020.1452
DOI: 10.3389/fneur.2020.00722
[4] “Conclusion”: In general, the “Conclusion” section of a paper is the part where the authors give a brief answer to the hypothesis they formulate, based on their research results. However, in this paper, the current "Conclusion" contains some things that should be discussed in the "Discussions" section. Therefore, I would suggest stating those things in the "Discussions" section and providing a brief answer to the hypothesis in the “Conclusion” section.
Author Response
Please see the attachment.
Point 1: Abstract” and “Introduction”: While the authors provide a detailed description of telemedicine, the hypotheses in this study are not clear. This makes it a little difficult for the reader to understand why this study was conducted. I would recommend that the authors provide a clear notation of what the hypothesis of this study is in the Abstract and Introduction.
Response 1: The decision to include such a detailed description of telemedicine in the introductory section was dictated by the need to provide even a reader less familiar with the subject with a complete representation of the particular area in which the clinical experience described was carried out.
The suggestion is however appropriate and useful to better clarify, also from a methodological point of view, the aims of the proposed study.
Point 2: Result”: I would recommend spelling out what “NHL” and “CLL” stand for.
Response 2: The text has been modified by integrating the extended definition of the acronyms previously reported.
Point 3: Discussion”: The authors discussed general cautions and other aspects of telemedicine. However, I believe that the discussion should focus upon what was done in this study and its results.
Response 3: the commentary highlighted an element of considerable importance that needed to be given greater importance; in acceptance of the observation, the work was integrated with a detailed description of the results obtained and the impact that telemedicine could have in daily clinical practice even in the post covid era. In particular, the results obtained highlighted the safety of the method.
Point 4: I found the results of this study itself remarkably interesting. I think it's worth discussing the fact that two of the 12 people who were low compliant had hearing problems. I think that is very interesting because while some of the previous papers have suggested that telemedicine may be difficult for people with hearing and vision problems, we don't have enough concrete data yet. I encourage authors to discuss these results in the discussion part. Please refer to the following papers:
DOI: 10.1001/jamaneurol.2020.1452
DOI: 10.3389/fneur.2020.00722
Response 4: The suggestion is interesting, but given the extreme complexity of the subject, deepening it in the article would lead to its excessive lengthening; it is undoubtedly an issue that deserves to be analysed in future research specifically concerning this topic.
Point 5: Conclusion”: In general, the “Conclusion” section of a paper is the part where the authors give a brief answer to the hypothesis they formulate, based on their research results. However, in this paper, the current "Conclusion" contains some things that should be discussed in the "Discussions" section. Therefore, I would suggest stating those things in the "Discussions" section and providing a brief answer to the hypothesis in the “Conclusion” section.
Response 5: We accepted the suggestion and some parts of the conclusion were included in the discussion.

Reviewer 3 Report
Overall, this manuscript is well-written. It described the experience of telemedicine during COVID-19.
- Please specify the objective of this study in introduction. The analysis and result of this study is quite limited.
- Do you have clinical characteristic data of patients
- There was no control in this study. How were the adverse events or events due to low compliance when using telemedicine, compared to face-to-face meeting.
- What is the limitations of this study
Author Response
Please see the attachment.
Point 1: Overall, this manuscript is well-written. It described the experience of telemedicine during COVID-19.
- Please specify the objective of this study in introduction. The analysis and result of this study is quite limited.
- Do you have clinical characteristic data of patients
- There was no control in this study. How were the adverse events or events due to low compliance when using telemedicine, compared to face-to-face meeting.
- What is the limitations of this study
Response 1: In response to point 1 we accepted the remark and subsequently integrated the opening section.
Referring to the other observations we have not included the characteristics of the patient data in order not to overload the article but it may be subject to subsequent publication; there are no known comparison data with which to compare ourselves. The lack of other uni-centric published experiences is the limit of the study, of which we are aware.

Round 2
Reviewer 1 Report
The comments have been addressed.
Reviewer 3 Report
all of my comments have been addressed. If not, they acknoweledged them as limitation of the study.